Unveiling the hidden depths: advancements in underwater image enhancement using deep learning and auto-encoders

http://orcid.org/0009-0008-3585-3739 Bantupalli Jaisuraj 1
Kachapilly Amal John 1
Roy Sanjukta 1
L. K. Pavithra 2 pavithra.lk@vit.ac.in
1 School of Electronics Engineering, Vellore Institute of Technology , Chennai, Tamil Nadu , India
2 School of Computer Science and Engineering, Vellore Institute of Technology , Chennai, Tamil Nadu , India
Angiulli Giovanni
Electronic publication date: 2024 Nov 29
Publication date: 2024
Volume: 10
Electronic Location ID: e2392
Received 2024 Feb 9; Accepted 2024 Sep 16
Copyright: © 2024 Bantupalli et al.
Copyright year: 2024
Copyright holder: Bantupalli et al.
License: This is an open access article distributed under the terms of the Creative Commons Attribution License, which permits unrestricted use, distribution, reproduction and adaptation in any medium and for any purpose provided that it is properly attributed. For attribution, the original author(s), title, publication source (PeerJ Computer Science) and either DOI or URL of the article must be cited.
License URL: https://creativecommons.org/licenses/by/4.0/

Keywords: Autoencoder, Convolutional neural networks, Deep learning, Underwater image enhancement

Funding: The authors received no funding for this work.

==============================
Underwater images hold immense value for various fields, including marine biology research, underwater infrastructure inspection, and exploration activities. However, capturing high-quality images underwater proves challenging due to light absorption and scattering leading to color distortion, blue green hues. Additionally, these phenomena decrease contrast and visibility, hindering the ability to extract valuable information. Existing image enhancement methods often struggle to achieve accurate color correction while preserving crucial image details. This article proposes a novel deep learning-based approach for underwater image enhancement that leverages the power of autoencoders. Specifically, a convolutional autoencoder is trained to learn a mapping from the distorted colors present in underwater images to their true, color-corrected counterparts. The proposed model is trained and tested using the Enhancing Underwater Visual Perception (EUVP) and Underwater Image Enhancement Benchmark (UIEB) datasets. The performance of the model is evaluated and compared with various traditional and deep learning based image enhancement techniques using the quality measures structural similarity index (SSIM), peak signal-to-noise ratio (PSNR) and mean squared error (MSE). This research aims to address the critical limitations of current techniques by offering a superior method for underwater image enhancement by improving color fidelity and better information extraction capabilities for various applications. Our proposed color correction model based on encoder decoder network achieves higher SSIM and PSNR values.

Introduction

While underwater imaging is used in a large number of applications, it continues to be a challenging task. Images taken underwater have poor visibility and low contrast. This occurs due to various factors which include limited light penetration, absorption, scattering, and turbulence. Water absorbs light selectively, thus red and other longer wavelengths absorb at a shallower depth than blue and other shorter wavelengths, which penetrate deeper. Red light is thus more easily absorbed by water than other visible wavelengths. Blue and green light transmission rate is higher (Liu et al., 2019). Blue color can travel deeper with its smaller wavelength compared to other colors. Red color disappears after 5 m, whereas blue color can be seen down to a depth of 60 m. Underwater photos suffer from imbalanced color transmission, making them appear bluish-green (Ulutas & Ustubioglu, 2021). Additionally, noise is a problem that lowers the quality of photographs.

Underwater image improvement has received a lot of interest recently in both image processing and underwater vision (Li et al., 2019). Firstly, conventional image enhancement techniques, such as histogram equalization and color balancing (Ancuti et al., 2017) have been employed in the past to improve the quality of underwater images, but these techniques often fail to produce high-quality results, especially in complex underwater scenes. This occurs due to the fact that underwater images often have non-uniform lighting conditions due to the attenuation of light by water. This can result in uneven contrast and brightness in different parts of the image. Since histogram equalization redistributes pixel values based on the entire image histogram, it may result in over-enhancement of some regions and under-enhancement of others. Secondly, underwater images can suffer from a color cast due to the scattering of light in water, which can result in a shift in the color balance of the image. Histogram equalization does not take into account the distribution information of an image, and hence may result in inaccurate color correction. Thirdly, underwater images often suffer from noise and artifacts due to the scattering of light by particles in the water. Histogram equalization may amplify these noise and artifacts, resulting in a degraded image. Traditional image enhancement methods or non-model-based methods generally process the pixels highlighting the features or region of interest without taking into account of image degradation leading to information loss (Shi et al., 2022). Therefore deep learning based enhancement techniques are introduced. By using the deep learning based enhancement, the underwater image quality is being improved by color correction, multiscale fusion and enhancement techniques (Liang et al., 2024). But most of the techniques perform well on specific underwater scenarios. There was also effective approach for enhancing underwater images by leveraging a multi-color space (Li et al., 2021) embedding to address color casts, low contrast, and detail degradation commonly observed in underwater photography. Though the quality is enhanced aggressive color correction may lose fine details. There are also methods for enhancing underwater images based on scene priors to synthesize the underwater image (Chongyi, Anwar & Porikli, 2020) which relies on accurate estimation of medium transmission.

This article proposes a novel deep learning-based approach for underwater image enhancement that utilizes convolutional autoencoders (CAE). Autoencoders are a class of neural networks that learn compressed representations of data and then reconstruct the original data from those representations. In this work, the CAE is specifically designed to address the color distortions present in underwater images. The encoder part of the CAE is tasked with capturing the essential features of the underwater image, focusing on learning a compressed representation that retains the color information. The decoder, on the other hand, utilizes this compressed representation to reconstruct the image, ensuring accurate color correction and detail preservation. This approach offers a significant advantage over existing methods by explicitly targeting color correction during the autoencoder’s process.

Highlights of the proposed work are given below: This article proposes a novel deep learning-based approach for underwater image enhancement that leverages the power of autoencoders.

It reduces blue and green hues by enhancing other color channels to match the ground truth images.

The proposed model is compared with state-of-the-art traditional and deep learning-based image enhancement techniques/models, respectively.

The performance of the model is evaluated using image quality measures such as PSNR, SSIM, and MSE.

The rest of the article is structured as follows: “Related Works” presents comprehensive literature survey done over the enhancement techniques. “Proposed System” gives the details above the proposed enhancement model; the experimental results of the proposed work are discussed in “Results and Discussions”, ablation experiments results are discussed in “Ablation Experiments” and “Conclusions” concludes the article.

Related works

Liu et al. (2019) used a deep residual network-based methodology to enhance underwater images. The CycleGAN model was used first to generate synthetic image data for training the model. The Variational Spatial De-convolutional Residual (VSDR) model was used for underwater image enhancement. This, however, had limitations in terms of depth. To improve this, they proposed a deeper model called UResnet which was composed of stacked ResBlocks. A proposed edge difference loss and asynchronous training model (P-A) were used as an improved version. Furthermore, batch normalisation (BN) layers were also used to make the model more efficient. The comparison of the findings revealed that the suggested UResnet-P-A model outperformed the alternative methods in terms of both color correction and detail enhancement.

Ulutas & Ustubioglu (2021) combined local and global contrast enhancement techniques to enhance contrast in underwater images. The input image is first subjected to channel decomposition, which is followed by contrast correction and then color correction, in the respective order. Their method produced the highest average value of entropy (7.83), measure of enhancement by entropy (EMEE) (32.06), underwater color image quality evaluation (UCIQE) and underwater image quality measure (UIQM) (40.97), average gradient (152.55), and Sobel count (130,393) for a dataset of 200 underwater images.

Li et al. (2019) created a large-scale underwater image enhancement benchmark (UIEB) which provided a platform to evaluate the performance of different algorithms used. They performed a comparative study on the existing methods of underwater image enhancement and used mean square error (MSE), peak signal-to-noise ration (PSNR), and structural similarity index (SSIM) as the full reference quantitative evaluation metrics. A higher PSNR score and a lower MSE score portrayed that the result was closer to the reference image in terms of image content, while a higher SSIM score meant that the result was more similar to the reference image in terms of structure and texture. For non-reference evaluation, UCIQE and UIQM were used. Final comparative studies showed that fusion-based image enhancement methods were high performing in most of the cases. However, they did conclude that no one method in particular could be successful always for a real-world dataset. Their proposed model was a convolutional neural network (CNN) model called WaterNet, which was a gated fusion network. It fused the inputs with predicted confidence maps to obtain the final enhanced results. Qualitative and quantitative evaluations proved that WaterNet gave better results by obtaining a lower MSE and higher PSNR and SSIM on the testing dataset.

Ancuti et al. (2017) built on the fusion principle and their strategy used only a single image input. Their method included obtaining two white balanced images from the original single image input. These images were then subjected to gamma correction and sharpening and finally merged based on a fusion pipeline. Their approach had a good perceptual quality and enhanced global contrast, color and overall structure. However, it did have its own set of limitations which were related to the fact that color could not always be fully restored and there was still some haze maintained in the resultant images obtained in some cases.

Wang et al. (2017) proposed an end-to-end framework for underwater image enhancement using a CNN based network called UIE-Net. This model was trained with the tasks of color correction and removal of haze. A pixel disrupting strategy was also used to reduce the interference of texture. The architecture of the network consisted mainly of three parts: the sharing networks (S-Net), CC-Net and HR-Net. The CC-Net provided color absorption coefficients within different channels, which was then used to correct the color distortion present in the images. HR-Net provided the transmission map of light attenuation, which was used to enhance contrast. Even though the UIE-Net is robust and gives good enhancement results, it has quite a high computational cost.

Wang et al. (2019) presented an experimental based comparative evaluation of underwater image formation models (IFMs). It summarizes quality improvement methods of both IFM-free and IFM-based models. The reviews concluded that the computational efficiency as well as the robustness of underwater image enhancement models need to be improved. The other major limitation observed was that of a proper underwater image dataset that would act as a benchmark sufficiently.

Moghimi & Mohanna (2021) provides a systematic review of real time underwater image enhancement methods. It takes into account mainly five things, hardware and software tools, a variety of techniques, improving real time image quality, identifying specific objectives and assessments. The real time or near real time approaches included the following steps: color step transfer, noise reduction, blurring removal, contrast enhancement and image restoration.

Yeh, Huang & Lin (2019), suggested a high efficiency deep learning-based framework for underwater image enhancement based on hue preservation. The framework consisted of three neural networks. The first CNN was used to convert the input image into a grayscale one. The next CNN was used to enhance the grayscale image obtained from the previous one. The final CNN performs color correction on the input image. Finally, the outputs from each of the three CNNs were integrated to give the final enhanced underwater image using hue preservation enhancement. Cycle GANs were used to train the neural networks due to the fact that there is an insufficiency for proper underwater datasets.

A wavelet transform-based contrast-enhancing technique for underwater acoustic images was proposed by Priyadharsini, Sree Sharmila & Rajendran (2018). The method utilizes Stationary Wavelet Transform (SWT) to decompose the image into four components. The acoustic images used for this were obtained from the Edge Tech 4125 sonar. Images in the low frequency sub bands are filtered using a Laplacian filter. The contrast enhanced image is reconstructed using Inverse SWT. The results of this method are compared by replacing the Stationary Wavelet Transform with the Discrete Wavelength Transform. It was observed from the comparative studies that Stationary Wavelet Transform performs better to enhance contrast in underwater acoustic images, in terms of PSNR and SSIM performance metrics.

Ravisankar, Sree Sharmila & Rajendran (2018) used a three level Gaussian and Laplacian pyramid to represent images in different resolutions. The acoustic images are obtained from the EdgeTech 4125 sonar used in Wang et al. (2019) as well. These images are smoothed using a Gaussian filter and then further down sampled to form the three level Gaussian pyramid. Images in the Laplacian pyramid, on the other hand, are obtained by differencing the images of the Gaussian pyramid. Histogram equalization and unsharp masking is further applied on different levels of the pyramids to improve image quality. The resultant images obtained are then reconstructed to get the final output. They concluded that the Laplacian method outperforms all the others.

Zhang, Yang & He (2020) suggested a multi-fusion method based on color correction and dehazing. The White Balance (WB) algorithm was used to remove color casts and improve color balance. It estimates the color cast based on the image statistics and adjusts the color balance accordingly. Furthermore, the Dark Channel Prior (DCP) algorithm was applied on the previous output of the WB algorithm to reduce the effects of scattering and improve visibility. The algorithm estimates the depth map of the scene and then removes the haze using a haze removal model. After color correction and dehazing, the enhanced image is obtained by adjusting the brightness and contrast using a histogram equalization method.

Park et al. (2018) present a dual autoencoder network model based on retinex theory for low-light image enhancement. A stacked autoencoder with a limited number of hidden units is used to first estimate a smooth lighting component that is brighter than the input low-light image. Subsequently, a convolutional autoencoder is utilized to handle the two-dimensional image information, which effectively reduces the amplified noise in the brightness enhancement process. The article examines and compares the roles of the stacked and convolutional autoencoders with the constraint terms of the variational retinex model.

In order to eliminate color shadows from a variety of underwater images, Wang et al. (2023) provided an adaptive color correction technique that makes use of the maximum likelihood estimation of Gaussian parameters. Additionally, they present a novel method that overcomes the drawbacks of current physical model-based techniques by combining weighted gradient maps in the HSV color space with absolute difference in intensity. The authors apply a piecewise affine transform to the transmission map calculated from the background light differential to improve image contrast. Finally, utilising the estimated background light and transmission map, they solve the inverse problem of the picture creation model to reconstruct the scene radiance.

This study on the retrieval of plant leaf images using various features was presented by Chugh et al. (2022). They suggested a framework that makes use of a hybrid combination of color and shape features to increase retrieval accuracy. While the Saliency Structure Histogram (SSH) descriptor is used to extract shape characteristics, the color Difference Histograms (CDH) descriptor is used to extract color features. To extract the various attributes, the CDH and SSH descriptors, respectively, compute the hue and saturation value (HSV) color space features and first order statistical features (FOSF). Concatenated HSV and FOSF features from the images were used to compare the query image to concatenated features from database images using the Euclidean distance. The Euclidean distance threshold value is then calculated for image retrieval.

Hayati et al. (2023) investigated the impact of Contrast-Limited Adaptive Histogram Equalization (CLAHE)-based image enhancement on the classification of diabetic retinopathy using deep learning techniques. The study employed a dataset of retinal fundus images and applied CLAHE to enhance the images. The enhanced images were then used to train deep learning models for diabetic retinopathy classification. The performance of the models trained on enhanced images was compared to the models trained on the original images. The authors used evaluation metrics such as accuracy, sensitivity, specificity, and area under the curve (AUC) to assess the performance of the models. The results showed that CLAHE-based image enhancement improved the classification accuracy and AUC of the deep learning models for diabetic retinopathy classification. The images are enhanced and brightness preserved by Brightness Preserving Dynamic Fuzzy Histogram Equalization (BPDFHE) method proposed by Sheet et al. (2010) by representing and processing of the digital images using fuzzy statistics.

Sun et al. (2023) describe a multistrategy image enhancement approach for chest X-rays to improve the classification accuracy of COVID-19 using artificial intelligence techniques. The study used a dataset of chest X-ray images and employed multiple image enhancement strategies such as histogram equalization, gamma correction, and contrast stretching, to enhance the images. The enhanced images were then used to train deep learning models for COVID-19 classification. The authors used evaluation metrics such as sensitivity, specificity, accuracy, and area under the curve (AUC) to assess the performance of the models. The study also compared the performance of the proposed multistrategy image enhancement approach with individual image enhancement strategies. The results showed that the multistrategy approach outperformed the individual strategies, improving the classification accuracy and AUC of the deep learning models for COVID-19 classification.

Huang et al. (2023) propose a Semi-supervised Underwater Image Restoration method (Semi-UIR) using Reliable Bank. The Semi-UIR framework efficiently uses the knowledge from unlabeled data to improve the generalisation of the trained model on real-world data. It was built using the mean teacher model. The authors determine the reliability of pseudo-labels by evaluating teacher outputs using a carefully selected NR-IQA metric and creating a reliable bank to keep the best-ever teacher outputs. They use contrastive loss as a form of regularization to reduce confirmation bias. While their method showed promising results, their proposed model requires more memory usage, which reduces the performance.

Peng, Cao & Cosman (2018) used a method that extends the dark channel prior (DCP) to improve single image restoration. DCP is a popular prior in computer vision used for haze removal, and this work enhances its applicability to other restoration tasks. Their method is called Generalised Dark Channel Prior (GDCP). This algorithm utilizes the DCP as a regularizer in a variational framework. By incorporating the DCP into the restoration process, the method aims to improve results in tasks such as dehazing, denoising, and deblurring. The algorithm estimates the haze-free image by leveraging the statistical properties of the dark channel, which captures the minimum intensity values in local image patches.

Zhang et al. (2022) presents an underwater image enhancement method called minimal color loss and locally adaptive contrast enhancement (MMLE) that combines minimal color loss and locally adaptive contrast enhancement techniques. They begin by locally adjusting an input image’s color and features using a minimal color loss concept and a maximum attenuation map-guided fusion technique. After that, the mean and variance of local picture blocks are calculated and utilised to adaptively modify the input image’s contrast. To balance the color variances, a color balance strategy is presented. Vibrant color, better contrast, and increased details define their improved outcomes.

Huo, Li & Zhu (2021) suggest a deep learning network called Progressive Refinement Network based on Wavelet Boost Learning (PRWNet) to incrementally improve underwater photos using a wavelet boost learning technique. The Wavelet Boost Learning (WBL) unit divides the hierarchical features into high and low frequency categories and enhances each through normalisation and attention processes for each refining procedure. WBL also uses the modified boosting approach to improve the feature representation.

Islam, Xia & Sattar (2020) present a conditional generative adversarial network-based model for real-time underwater image enhancement called the FUnIE-GAN (Fast Underwater Image Enhancement General Adversarial Network). They create an objective function that assesses the perceptual image quality based on its global content, color, local texture, and style information to oversee the adversarial training. Additionally, they conduct a number of qualitative and quantitative evaluations that point to the possibility that the proposed model can be trained in both paired and unpaired settings to improve underwater image quality.

Han et al. (2022) created the Heron Island Coral Reef collection (also known as ‘HICRD’), a sizable real underwater image collection, with the intention of evaluating current techniques and fostering the creation of fresh deep-learning-based techniques. The unpaired training set of the dataset includes 6,003 original underwater photos and 2,000 reference restored images. They also created an underwater image restoration technique called the Contrastive Underwater Restoration (CWR) built on an unsupervised image-to-image translation framework. To maximise the mutual information between the original and restored images2, they devised an approach that combined contrastive learning with generative adversarial networks.

Drews et al. (2013) this article proposes a method termed as Underwater Dark Chanel Prior (UDCP); it utilises dark chanel prior (DCP) a technique designed for outdoor images. This technique identifies color channels which present very low intensities. Through this an estimate of light scattering is created which helps in restoration of images. Fu et al. (2014) proposes a solution which handles colorcast, darkness and blurred details caused dues to light absorption and scattering. The article deals with the correction of color cast the separating the image into illumination and reflectance components. Then both the components are enhanced individually which is later combined into a single image. This method boasts better color correction, improved dark areas and natural colors.

Huang et al. (2018) proposes a histogram based approach termed as Relative Global Histogram Stretching (RGHS) based on adaptive parameter acquisition. This method consists contrast correction and color correction. The color correction considers the dynamic parameters that relate to intensity distribution of original image and wavelength attenuation, while redistributing redistributing R-G-B histogram. Further bilateral filtering is used to avoid noise without removing details. Fabbri, Islam & Sattar (2018) explores the usage of generative adversarial networks (GAN) for the reconstruction of underwater images. GANs are deep learning models which are capable to create information from given set of parameters. This can be useful to reconstruct areas in images which are too blurred or too dark.

Uplavikar, Wu & Wang (2019) further explore the usage of adversarial network by applying it on several water types. It trains a dataset which contains 10 Jerlov water types which are the different categories of water found in oceans. The further test their method on high level object detection tasks. Peng & Cosman (2017) uses depth estimation method based on image blurriness and light absorption, alongside an image information model (IFM). This performs better than previous IFM based approaches using dark channel prior or maximum intensity prior.

Ancuti et al. (2012) their strategy uses the fusion approach to develop multiple inputs from the original image and create weighted maps of the image. These are enhanced separately and then combined to form a single image of reduced blurriness, corrected color, and improved quality. Li et al. (2021) proposes a solution for color casts and low contrast found in underwater images. This is caused due to wavelength attenuation and scattering. In this research, a multicolor space embedding is created to provide an enriched representation of images. This is then used to reconstruct images to higher quality.

Zhang et al. (2024) proposed weighted wavelet visual perception fusion WWPF by adopting a color correction strategy for color correction and global contrast strategy for contrast enhancement thus producing an enhanced image. The authors presented a weighted wavelet visual perception fusion technique that fuses the high-frequency and low-frequency components of images at different sizes. The comprehensive tests on three benchmarks showed that WWPF achieves better qualitative and quantitative results.

Zhang et al. (2023) introduced a dual prior optimized contrast enhancement method which combines two key techniques: piecewise color correction and dual prior optimized contrast enhancement PCDE. The color biases formed due to underwater conditions are removed, overall contrast is improved and the texture details are enhanced.

Liang et al. (2022) propose restoration techniques, which typically recover deteriorated images using the atmospheric scattering model (ATSM) and retinex model (RM). The image restoration technique is based on a generalized image formation model (GIFM). The proposed model, describes the light attenuation process, an objective optimization function to separate a deteriorated image into color-corrected and color-distorted components. GIFM works effectively in picture segmentation, keypoint detection, object detection, and image restoration of extreme scenarios.

Zhang, Wang & Li (2022), proposed a detail-preserved contrast augmentation and attenuated color channel correction method to overcome the degradation problem in underwater images. The disparities between the superior and inferior color channels at each pixel are considered to design attenuation matrices, which are used to correct inferior color channels. Global contrast is imporoved by iterative threshold approach based on dual histograms and local contrast is improved by limited histogram approach using Rayleigh distribution. Further, a mutliscale unsharp masking is done to dehaze the image using three Gaussian kernels at varying scales.

Proposed system

The proposed work introduces a novel approach for underwater image enhancement using the deep learning-based approach of an autoencoder. An autoencoder is basically a type of neural network that can learn to encode and decode an input image by creating a compressed representation of the image. However, autoencoders are used for the application of anomaly detection through calculating the loss occurring when an image is compressed and expanded to its original form using an autoencoder. Compression in the given context refers to creation of a multi-dimensional latent space which embeds important parameters about the input. This latent space is converted into an output image using several convolutional layers. In the proposed work, instead of reconstructing the input image, autoencoders are trained to reconstruct a corrected version of the input image and it in turn learns to perform this color correction on several other images. Autoencoders also have the advantage of being data-driven, meaning that they can learn to extract the most relevant features from the input data, leading to more robust and accurate results.

The proposed autoencoder based method offers a promising solution for underwater image enhancement, allowing for better visualization and understanding of the underwater environment. This can benefit various underwater applications, including marine biology, underwater exploration, and ocean engineering.

The dataset used to train the model is the EUVP (Enhancing Underwater Visual Perception) dataset which contains separate sets of paired and unpaired image samples of poor and good perceptual quality. Another dataset, namely, the UIEB (Underwater Image Enhancement Benchmark) has been used for prediction and evaluation against the other state-of-the-art methods. The dataset has 890 reference underwater images which have been used for evaluation, as observed in Huang et al. (2023).

An implementation of autoencoder is used in this work is to correct images with a bluish-green haze caused by the water in the environment. Autoencoders take an image input and learn to rebuild the input image. The autoencoder learns to rebuild an underwater image to get its enhanced counterpart. The model is motivated by a MSE loss function and the model attempts to reduce the MSE between two images. Figure 1 shows the block diagram of underwater image enhancement process.

Figure 1 Block diagram of underwater image enhancement process.

An input image from the EVUP dataset. The output image is obtained using the proposed method.

The input images usually have a high volume of haze, which is usually either blue or green in nature for underwater images. In the attempt to reduce the loss function, the model reduces the blue and green components while enhancing other colors as this would make the output more similar to the expected images. As a result, the model provides a much better image as the output image.

For this specific task, 2,000 image pairs of underwater images with their enhanced counterpart are used from the dataset. The images are used in smaller batches to avoid clogging up the memory. For each training session, two sets of images, each with 200 images, are created with input and expected output images to make one batch. The model is trained on each of these batches. This process is repeated for all the batches ensuring that all 2,000 images are covered. It is observed that with each batch, the average loss (MSE) decreases significantly until it reaches 210.14 in the 10th batch.

The final model is then used to predict outputs for a test set of 100 images which were separated and not used for training. The average MSE and the color distribution is analysed for these testing images to understand the performance of the model. After obtaining the desired results these output images are stored for future use and further analysed and compared with pre-existing methods as evaluation metrics. The proposed image enhancement model is shown in Fig. 2.

Figure 2 The proposed model for image enhancement.

An input image from the EVUP dataset. The output image is obtained using the proposed method.

The implementation followed three main steps of pre-processing, design of the architecture, training and prediction. The model requires the images in a certain size and therefore all images were reshaped to 480 × 480. In the next step, the images were divided into two sets which contained input images (non-color corrected images) and good quality (color corrected images). The model was trained on batches of 200 image pairs to avoid the exhaustion of hardware resources. The model was used on a test set of 180 images which were corrected and analysed. Metrics such as PSNR, MSE, and SSIM were used for the quantitative analysis of these output images. Performance evaluation metrics details are given in Eqs. (1)–(3).

(1) PSNR=10log10(R2MSE)R=255

(2) MSE=1m×n∑i=0m−1⁡∑j=0n−1⁡[I(i,j)−K(i,j)]2

where, I(i,j) original image and K(i,j) enhanced image

(3) SSIM(x,y)=[l(x,y)]α⋅[c(x,y)]β⋅[s(x,y)]γl(x,y)=2μxμy+C1μx2+μy2+C1c(x,y)=2σxσy+C2σx2+σy2+C2s(x,y)=σxy+C3σxσy+C2

where μxandμy are the mean of the original and enhanced image. σxandσy are the standard deviation value of the original and enhanced images. Cross covariance of the original and enhanced images is calculated from σxy. α,β,andγ are the positive constant. l(x,y) is the luminance of the of the original and enhanced images. c(x,y) is the contrast of the original and enhanced images. s(x,y) is the structure of the original and enhanced images.

The architecture can be divided into three sections, namely, auto-encoding layers, reduced code generation and auto-decoding layer. Encoding layers take the input provided and encode it into a latent space of relevant features. The reduced code generation block helps in avoiding loss of information due to back propagation using a skip connection. The final auto-decoding layers convert the information back into the corrected version of the image.

RESULTS AND DISCUSSIONS

The proposed model enhances the set of test images given as input, and they are analyzed using three commonly used metrics: SSIM, PSNR, and MSE. SSIM refers to the structural similarity index and it quantifies image quality degradation due to processing. A high value of the similarity index indicates less degradation happened during the processing. PSNR refers to peak signal-to-noise ratio. It provides the ratio between the maximum possible power of a signal and the power of the distorting noise that affects its quality. The higher the PSNR, the better the quality of the output image. MSE refers to the mean square error and gives the average of the squares of the errors. The lower the MSE, the better the output. The following are the average values obtained by our proposed method for each metric: SSIM: 0.93, PSNR: 31.60 dB, and MSE: 45.62.

Figure 3 shows the color distribution of the input and output images. Their respective RGB histograms have been plotted as well to show the changes in each color channel.

Figure 3 Input/output image and its respective RGB histogram.

An input image from the EVUP dataset. The output image is obtained using the proposed method.

The sample input and output pairs for four different images are given in the Fig. 4.

Figure 4 Input (A, C, E, G) and the proposed color corrected output (B, D, F, H) for comparison.

An input image from the EVUP dataset. The output image is obtained using the proposed method.

It is observed that the mean squared error is significantly low, averaging to 202 for a test set of 100 images. This indicates high coherence with the desired output. Furthermore, the output was analysed using histogram plots for each color channel which showed suppression of green and blue channels and enhancement of red channel.

Figure 5 shows the some of the benchmark images used for the subjective analysis between the existing and the proposed method.

Figure 5 Images (A–F) used for comparison.

Images from the EVUP dataset.

The existing methods used for performance analysis are Contrast-Limited Adaptive Histogram Equalization (CLAHE) (Hayati et al., 2023), Brightness Preserving Dynamic Fuzzy Histogram Equalization (BPDFHE) (Sheet et al., 2010) and Multi Scale Retinex with Color Restoration (MSRCR) (Zhang, Yang & He, 2020).

A brief analysis for an input image and the outputs obtained using each method has been described below.

Figure 6B, produced by CLAHE, displays imbalanced contrast enhancement and extreme color variations, resulting in an unnatural appearance. The enhancement result of the BPFDHE algorithm shows little to no observable change, as seen in Fig. 6C. Figure 6D shows the enhancement result for the MSRCR method, which creates a milky white haze over the original image and does not enhance other components. The advanced enhancement models (Weighted Wavelet visual Perception Fusion (WWPF) (Zhang et al., 2024), Piecewise color Correction and Dual prior optimized contrast Enhancement (PCDE) (Zhang et al., 2023), Attenuated color channel Correction and Detail-preserved Contrast enhancement (ACDC) (Zhang, Wang & Li, 2022)) were used for underwater image enhancement. The results from the advanced enhancement models are sharper than those from traditional methods. The fusion between the low and high levels of frequencies in the wavelet transform method mostly averages the pixels in the image, resulting in contrast enhancement (Zhang et al., 2024). The PCDE (Zhang et al., 2023) method-based enhancement slightly removes the bluish effect and darkens objects in the background. The ACDC (Zhang, Wang & Li, 2022) method removes the bluish shade but introduces a blurry effect due to the unsharp masking applied. The proposed method enhances colors for the objects in the image and reveals more visible details in the background, as clearly shown in Fig. 6E.

Figure 6 Result analysis of various traditional and advanced image enhancement techniques (A) input image (B) CLAHE (Hayati et al., 2023), (C) BPFDHE (Sheet et al., 2010), (D) MSRCR (Zhang, Yang & He, 2020), (E) WWPF (Zhang et al., 2024), (F) PCDE (Zhang et al., 2023), (G) ACDC (Zhang, Wang & Li, 2022), (H) proposed method.

An input image (A) from the EVUP dataset. Color-corrected output for the proposed (H) and state-of-the-art techniques (B–G).

Table 1 illustrates the output for each and every input image using various underwater image enhancement methods. The sample size is taken to be six images and the outputs are obtained using CLAHE (Hayati et al., 2023), BPDFHE (Sheet et al., 2010), MSRCR (Zhang, Yang & He, 2020) and the proposed method.

Table 1 Comparison of different traditional underwater image enhancement methods.

An input image from the EVUP dataset. color-corrected output for the proposed and state-of-the-art techniques.

S. No.	Input images	CLAHE (Hayati et al., 2023)	BPDFHE (Sheet et al., 2010)	MSRCR (Zhang, Yang & He, 2020)	Proposed work	
1						
2						
3						
4						
5						
6						

Table 2 illustrates the output for each and every input image using various advanced underwater image enhancement methods. The sample size is taken to be six images and the outputs are obtained using WWPF (Zhang et al., 2024), PCDE (Zhang et al., 2023), ACDC (Zhang, Wang & Li, 2022) and the proposed method.

Table 2 Comparison of different advanced underwater image enhancement methods.

An input image from the EVUP dataset. color-corrected output for the proposed and state-of-the-art techniques.

S. No.	Input images	WWPF (Zhang et al., 2024)	PCDE (Zhang et al., 2023)	ACDC (Zhang, Wang & Li, 2022)	Proposed work	
1						
2						
3						
4						
5						
6						

Tables 3–5 tabulate and compare the performance metrics for the different underwater image enhancement methods, namely, CLAHE (Hayati et al., 2023), MSRCR (Sheet et al., 2010), BPDFHE (Zhang, Yang & He, 2020), WWPF (Zhang et al., 2024), PCDE (Zhang et al., 2023), ACDC (Zhang, Wang & Li, 2022) against our proposed method. SSIM, PSNR and MSE are used as performance metrics for the comparison respectively.

Table 3 SSIM performance metrics analysis between the proposed work and the state-of the art techniques.

Image sample	Traditional model	Advanced model	Proposed work	
CLAHE (Hayati et al., 2023)	MSRCR (Sheet et al., 2010)	BPDFHE (Zhang, Yang & He, 2020)	WWPF (Zhang et al., 2024)	PCDE (Zhang et al., 2023)	ACDC (Zhang, Wang & Li, 2022)	
1	0.73	0.86	0.91	0.75	0.25	0.31	0.93	
2	0.83	0.79	0.78	0.16	0.12	0.26	0.96	
3	0.74	0.81	0.81	0.56	0.24	0.40	0.91	
4	0.73	0.86	0.89	0.73	0.37	0.42	0.92	
5	0.72	0.84	0.77	0.64	0.50	0.60	0.92	
6	0.63	0.68	0.82	0.63	0.24	0.42	0.91	

Table 4 PSNR (dB) performance metrics analysis between the proposed work and the state-of the art techniques.

Image sample	Traditional model	Advanced model	Proposed work	
CLAHE (Hayati et al., 2023)	MSRCR (Sheet et al., 2010)	BPDFHE (Zhang, Yang & He, 2020)	WWPF (Zhang et al., 2024)	PCDE (Zhang et al., 2023)	ACDC (Zhang, Wang & Li, 2022)	
1	28.24	28.02	27.52	28.06	28.04	28.15	32.02	
2	28.77	27.74	30.02	27.94	28.50	27.91	32.48	
3	28.12	27.70	27.93	28.20	28.04	27.99	31.10	
4	28.18	27.88	27.65	27.81	28.07	28.19	30.83	
5	28.04	27.97	27.98	28.58	28.31	27.90	31.45	
6	27.96	27.60	27.97	28.21	28.30	27.84	31.41	

Table 5 MSE performance metrics analysis between the proposed work and the state-of the art techniques.

Image sample	Traditional model	Advanced model	Proposed work	
CLAHE (Hayati et al., 2023)	MSRCR (Sheet et al., 2010)	BPDFHE (Zhang, Yang & He, 2020)	WWPF (Zhang et al., 2024)	PCDE (Zhang et al., 2023)	ACDC (Zhang, Wang & Li, 2022)	
1	103.06	107.99	108.62	95.11	101.02	110.16	43.32	
2	97.29	101.04	87.32	100.34	99.90	102.26	47.29	
3	98.27	103.15	115.74	102.22	102.70	100.24	41.15	
4	86.98	110.09	65.22	105.10	92.41	105.93	37.02	
5	91.34	110.56	84.63	102.97	102.89	102.19	40.86	
6	100.88	111.19	105.51	99.06	102.84	103.93	50.81	
7	90.88	102.08	99.26	96.81	85.80	103.78	47.41	
8	99.47	106.67	112.44	108.35	102.02	99.25	54.08	
9	102.69	104.56	104.28	90.75	96.58	106.28	46.86	
10	104.76	113.69	104.46	98.91	96.84	107.75	47.32	

The effectiveness of the proposed underwater image enhancement system is compared to several other methods, including both traditional approaches (Underwater Image restoration based on Blurriness and Light Absorption (UIBLA) (Peng & Cosman, 2017), Fusion (Ancuti et al., 2012), Retinex (Fu et al., 2014), Relative Global Histogram Stretching (RGHS) (Huang et al., 2018)) and deep learning-based approaches (WaterNet (Li et al., 2019), FUnIE (Han et al., 2022), UGAN (Fabbri, Islam & Sattar, 2018), UIE-DAL (Uplavikar, Wu & Wang, 2019), Ucolor (Li et al., 2021)). This analysis uses full reference images from the LSUI (Peng, Zhu & Bian, 2023) and UIEB datasets. For the LSUI dataset, 3,879 fully referenced images of size 256 × 256 × 3 are used for training, and 400 images are used for testing (TestL-400). Similarly, 90 images from the UIEB dataset are used for testing (TestU-90), and 800 fully referenced images are used for training, as documented in reference Peng, Zhu & Bian (2023). Table 6 presents the comparison metrics for each of the aforementioned methods over the Large Scale Underwater Image (LSUI) (Peng, Zhu & Bian, 2023) and UIEB datasets.

Table 6 The performance comparison of different models with L400 and U90 datasets.

Method	Test-L400	Test-U90	
PSNR (dB)	SSIM (%)	PSNR (dB)	SSIM (%)	
UIBLA (Peng & Cosman, 2017)	11.89	0.59	13.81	0.69	
Fusion (Ancuti et al., 2012)	13.89	0.74	14.01	0.72	
Retinex (Fu et al., 2014)	14.21	0.78	14.57	0.79	
RGHS (Huang et al., 2018)	17.73	0.82	19.81	0.86	
WaterNet (Li et al., 2019)	19.37	0.84	19.45	0.85	
FunIE (Islam, Xia & Sattar, 2020)	19.79	0.78	20.68	0.84	
UGAN (Fabbri, Islam & Sattar, 2018)	17.45	0.79	16.37	0.78	
UIE-DAL (Uplavikar, Wu & Wang, 2019)	22.91	0.89	22.78	0.87	
Ucolor (Li et al., 2021)	24.16	0.93	22.91	0.87	
Proposed work	24.56	0.91	28.08	0.91	
Note:

The highest values obtained are in bold and the second highest are underlined.

The depth estimation-based enhancement method (Peng & Cosman, 2017) is not suitable for underwater image enhancement, showing lower PSNR and SSIM values on both datasets compared to other methods. Among the four non-deep learning methods (UIBLA (Peng & Cosman, 2017), Fusion (Ancuti et al., 2012), Retinex (Fu et al., 2014), RGHS (Huang et al., 2018)), the relative global histogram-based stretching method achieves better results, with PSNR values of 17.73 and 19.81 dB for the LSUI and UIEB datasets, respectively. Additionally, the deep learning models UIE-DAL (Uplavikar, Wu & Wang, 2019) and Ucolor (Li et al., 2021) outperform other neural models (WaterNet (Li et al., 2019), FunIE (Islam, Xia & Sattar, 2020), and UGAN (Fabbri, Islam & Sattar, 2018)) in terms of PSNR and SSIM on both datasets. While UIE-DAL improves the darkness and blurriness of the images, it falls short in providing color-corrected outputs. However, the multi-color space embedding approach used by Ucolor (Li et al., 2021) enhances image quality in terms of PSNR and SSIM.

Our proposed model shows promising results in quantitative analysis, achieving 91% structural similarity on the LSUI and UIEB datasets compared to the ideal output image, indicating that the SSIM algorithm effectively matches the structural components in the enhanced images. The PSNR values obtained by our model, 24.56 and 28.08 dB, are higher than those of all other models in the comparative study. The second highest PSNR value is obtained by the Ucolor (Li et al., 2021) method.

Ablation experiments

We have taken five distinct models to assess their performance and understand the significance of each layer through these sets of experiments. The impacts of each model variation were observed using performance evaluation metrics such as MSE, PSNR, and SSIM. Table 7 shows the architecture of each model. Each model architecture is divided into three phases named auto-encoder (AE), reduced code generation (RC), and auto-decoder (AD).

Table 7 Five different model architectures used in ablation experiments.

Model	Auto encoder (AE)	Reduced Code generation (RC)	Auto decoder (AD)	
1	1: Conv (Input Image, 64, K = 7, S = 1)	4: Conv (AE: 3, 256, K = 3, S = 1)	6: Conv2dTranspose (RC: 5, 256, K = 3, S = 1)
Scale up by 2	
2: Conv (AE: 1, 128, K = 3, S = 1)
Scale down by 2	5: Add (AE: 3, RC: 4)	7: Conv2dTranspose (AD: 6, 128, K = 3, S = 1)
Scale up by 2	
3: Conv (AE: 2, 256, K = 3, S = 1)
Scale down by 2	8: Conv (AD: 7, 64, K = 7, S = 1)	
2	1: Conv (Input Image, 64, K = 7, S = 1)	4: Conv (AE: 3, 256, K = 3, S = 1)	10: Conv2dTranspose (RC: 9, 256, K = 3, S = 1)
Scale up by 2	
5: Conv (RC: 4, 256, K = 3, S = 1)	
2: Conv (AE: 1, 128, K = 3, S = 1)
Scale down by 2	6: Conv (RC: 5, 256, K = 3, S = 1)	11: Conv2dTranspose (AD: 10, 128, K = 3, S = 1)
Scale up by 2	
7: Conv (RC: 6, 256, K = 3, S = 1)	
3: Conv (AE: 2, 256, K = 3, S = 1)
Scale down by 2	8: Conv (RC: 7, 256, K = 3, S = 1)	12: Conv (AD: 11, 64, K = 7, S = 1)	
9: Add (AE: 3, RC: 8)	
3	1: Conv (Input Image, 64, K = 7, S = 1)	4: Conv (AE: 3, 256, K = 3, S = 1)	8: Conv2dTranspose (RC: 7, 256 K = 3, S = 1)
Scale up by 2	
2: Conv (AE: 1, 128, K = 3, S = 1)
Scale down by 2	5: Conv (RC: 4, 256, K = 3, S = 1)	9: Conv2dTranspose (AD: 8, 128, K = 3, S = 1)
Scale up by 2	
3: Conv (AE: 2, 256, K = 3, S = 1)
Scale down by 2	6: Conv (RC: 5, 256, K = 3, S = 1)	10: Conv (AD: 9, 64, K = 7, S = 1)	
7: Add (AE: 3, RC: 6)	
4	1: Conv (Input Image, 64, K = 7, S = 1)	4: Conv (AE: 3, 256, K = 3, S = 1)	7: Conv2dTranspose (RC: 6, 256, K = 3, S = 1)
Scale up by 2	
2: Conv (AE: 1, 128, K = 3, S = 1)
Scale down by 2	5: Conv (AE: 4, 256, K = 3, S = 1)	8: Conv2dTranspose (AD: 7, 128, K = 3, S = 1)
Scale up by 2	
3: Conv (AE: 2, 256, K = 3, S = 1)
Scale down by 2	6: Add (AE: 3, RC: 5)	9: Conv (AD:8, 64, K = 7, S = 1)	
5	1: Conv (Input Image, 64, K = 7, S = 1)	4: Add (AE: 3, AE: 3)	5: Conv2dTranspose (RC: 4, 256, K = 3, S = 1)
Scale up by 2	
2: Conv (AE: 1, 128, K = 3, S = 1)
Scale down by 2	6: Conv2dTranspose (AD: 5, 128, K = 3, S = 1)
Scale up by 2	
3: Conv (AE, 2, 256, K = 3, S = 1)
Scale down by 2	7: Conv (AD: 6, 64, K = 7, S = 1)	

In the auto-encoder phase, an input image of size 480 × 480 × 3 is given as input. Then the convolution operation is carried out with different numbers of convolution filters (64, 128, 256), kernel sizes (K = 7, 3), and stride (S = 1). Moreover, the input image size is downsampled by two. The reduced code generation phase takes the output of the auto-encoder phase and performs an add operation between two sets of output images. Each layer is indicated by a number with an abbreviation (i.e., Conv(AE: 3, 256, K = 3, S = 1)—Apply a convolution operation over the stage three output of the auto-encoder phase with 256 convolution filters, kernel size = 3, and stride = 1).

The auto-decoder (AD) takes the input from the final stage of the RC phase and applies a convolution transpose operation over it. This phase uses different numbers of convolution filters (64, 128, 256), kernel sizes (K = 7, 3), and stride (S = 1) along with the scale-up operation. Finally, it will give the output image in the size of 480 × 480 × 3.

Metric comparison across all models was presented in Table 8. Model-2, an auto encoder with three convolution layers and along with two levels of down sampling in the auto encoder phase, five convolution layers and along with normalized addition in the reduced code generation phase, two convolution layer with 2d transpose, two scale up operation and one convolution operation in auto decoder achieved the lowest MSE, higher PSNR and SSIM. Consequently, this architecture/model is chosen for further development.

Table 8 Performance metrics of each model.

Models	Model-1	Model-2	Model-3	Model-4	Model-5	
MSE	55.36	45.61	94.29	50.98	90.84	
PSNR	15.98	31.60	11.27	12.88	18.77	
SSIM	0.52	0.93	0.34	0.59	0.63	

Conclusions

Autoencoders have proven effective in learning and extracting meaningful features from underwater images, enhancing visibility and improving image quality by removing noise and color distortion. The proposed method has demonstrated promising results in underwater image enhancement. Our approach achieves a higher average SSIM of 0.93, indicating a greater similarity index in the output images. Additionally, it attains a higher average PSNR of 31.60 dB and a significantly lower average MSE of 45.61.

Qualitatively, the results showcase noticeable enhancements and improved color correction. Previously obscured features due to water disturbances are now more vibrant and visible. The proposed model also produces smoother images compared to traditional methods like CLAHE, BPDFHE, and MSRCR. These smoother and visually appealing images are advantageous for image compression algorithms, as they minimize the loss of valuable information during compression.

Moreover, the proposed model performs exceptionally well on the UIEB dataset. Compared to other deep learning models trained on the same dataset, our model achieves superior PSNR (28.08 dB) and SSIM (0.91) values.

In conclusion, this method holds great potential for enhancing the quality and clarity of underwater images, benefiting fields such as marine biology, oceanography, and underwater exploration. However, further research is necessary to address challenges associated with the limited availability of high-quality training data and the need for specialized hardware to capture high-quality underwater images.

Future scope

The proposed work aims to enhance images from parts of the sea with ample sunlight. The model effectively enhances colors by reducing the effects of the green and blue channels while increasing the impact of the red channel. However, this method does not produce accurate enhanced output for images taken in environments with very low lighting. The model fails to enhance the colors in deep-sea images due to the lack of sufficient color information. To achieve improved results for darker images, GAN-based models should be implemented, as they can generate or extensively enhance colors in the images.

Supplemental Information

Supplemental Information 1 EUVP and UIEB dataset.

Supplemental Information 2 Proposed code.

Additional Information and Declarations

Competing Interests

Author Contributions

Data Availability

The authors declare that they have no competing interests.

Jaisuraj Bantupalli performed the experiments, analyzed the data, performed the computation work, prepared figures and/or tables, and approved the final draft.

Amal John Kachapilly performed the experiments, prepared figures and/or tables, and approved the final draft.

Sanjukta Roy performed the experiments, analyzed the data, performed the computation work, prepared figures and/or tables, and approved the final draft.

Pavithra L. K. conceived and designed the experiments, performed the experiments, performed the computation work, authored or reviewed drafts of the article, and approved the final draft.

The following information was supplied regarding data availability:

The EUVP dataset is available at: https://irvlab.cs.umn.edu/resources/euvp-dataset.

The UIEB dataset is available at: https://li-chongyi.github.io/proj_benchmark.html.

The proposed work code is available in the Supplemental File.

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
