# Peer review of "Unveiling the hidden depths: advancements in underwater image enhancement using deep learning and auto-encoders"

_PeerJ Computer Science, doi:10.7717/peerj-cs.2392_

## Round 0.1 · original submission · Major Revisions

Dear Authors,

Your paper has been revised. Given the reviewers' opinions, your paper needs major revisions. The authors must face the following points before resubmitting their work.

i) The paper's structure must be improved, and the methodological review must identify critical issues. Authors must clearly explain the methodological innovation. Furthermore, the author needs to re-condense the abstract; the background of the abstract needs to be more varied. The abstract first identifies the challenging problem faced, then the specific solution, and finally the experimental results, keeping the abstract between 150 and 200 words

ii) The experimental analysis must be better organized. The experimental comparison employs outdated methods and needs more contemporary comparative approaches, which undermines the validation of the proposed method's effectiveness. Furthermore, the article needs more comparative methods for comparative tests and needs comparison with the latest techniques. Finally, the authors need to add ablation experiments.

Iii) The article's figures are unclear; authors are requested to provide high-resolution figures.

**Language Note:** The review process has identified that the English language must be improved. PeerJ can provide language editing services - please contact us at [email protected] for pricing (be sure to provide your manuscript number and title). Alternatively, you should make your own arrangements to improve the language quality and provide details in your response letter. – PeerJ Staff

Reviewer 1 ·

Basic reporting

The article presents a deep learning-based autoencoder method for enhancing underwater images. This approach utilizes autoencoders for image compression and decompression while simultaneously learning color correction in the underwater environment. Additionally, it trains the autoencoder to extract the most relevant features from underwater images and corrects these features to enhance image quality. The effectiveness of this method is verified through experiments, although certain issues are identified simultaneously:

1. For the summary, it should be straightforward. The author needs to re-condense the abstract, the background of the abstract is too redundant. The abstract first identifies the challenging problem faced, then the specific solution, and finally the experimental results, keeping the abstract between 150 and 200 words.

2. To make the paper more clearly organized, in the "Introduction" part, the author needs to summarize the innovation points of the method proposed in this paper and list them in three parts.

3. In the Introduction, the framework listed by the author is not clear enough, the content structure is not comprehensive enough, and the classification is not clear enough. (May refer to and quote: "Weighted
Wavelet Visual Perception Fusion" "piecewise color correction and dual prior optimized contrast enhancement" and "An Image Restoration Method With Generalized Image Formation Model").

4. In the "Related work" section, the introduction of others' methods is too redundant and the quoted methods are not novel enough (for deep learning, you can refer to and cite: "Underwater Image Quality Improvement via Color, Detail, and Contrast Restoration", "Medium Transmission-Guided Multi-Color Space Embedding" and "Underwater Scene Prior Inspired ").

5. The flow chart needs to be redrawn, and some visual auxiliary pictures should be added accordingly. The enhanced result image needs to be added to the general flow chart, and the step introduction of the entire algorithm workflow needs to be added to the lower part of the flow chart. Please ask the author to optimize it.

6. The figures in the article are not clear; authors are requested to provide high-resolution figures.

7. The article has too few comparative methods for comparative tests and lacks comparison with the latest methods. (The following methods can be referred to: "Minimal Color Loss and Locally Adaptive Contrast Enhancement", "attenuated color channel correction and detail preserved contrast enhancement" and "hyper-laplacian reflectance priors").

8. The article lacks ablation experiments; the authors need to add ablation experiments.

Experimental design

As above

Validity of the findings

As above

Additional comments

No comment

Reviewer 2 ·

Basic reporting

The paper's structure is too chaotic, and the methodological review fails to identify key issues. The methodological innovation is weak, and the experimental analysis is rather disorganized. It is recommended that the authors re-optimize the paper's structure and carefully revise the content.
The experimental comparison employs outdated methods and lacks contemporary comparative approaches, which undermines the validation of the proposed method's effectiveness. Consequently, I am unable to assess the workload of the paper.

Experimental design

The experimental comparison employs outdated methods and lacks contemporary comparative approaches, which undermines the validation of the proposed method's effectiveness. Consequently, I am unable to assess the workload of the paper.

Validity of the findings

The experimental comparison employs outdated methods and lacks contemporary comparative approaches, which undermines the validation of the proposed method's effectiveness. Consequently, I am unable to assess the workload of the paper.

Additional comments

The experimental comparison employs outdated methods and lacks contemporary comparative approaches, which undermines the validation of the proposed method's effectiveness. Consequently, I am unable to assess the workload of the paper.

---

## Round 0.2 · accepted · Accept

Dear Authors,
Your paper has been accepted for publication in PeerJ Computer Science. Thank you for your fine contribution.

Reviewer 1 ·

Basic reporting

no comment

Experimental design

no comment

Validity of the findings

no comment

Additional comments

All my concerns have been well addressed. The manuscript can be recommended for publication.